# Calcium Phosphate Loaded Biopolymer Composites—A Comprehensive Review on the Most Recent Progress and Promising Trends

**Monika Furko \*, Katalin Balázsi** and **Csaba Balázsi**

Centre for Energy Research, ELKH, HU1121 Konkoly-Thege M. rd 29-33, H-1121 Budapest, Hungary
\* Correspondence: furko.monika@ek-cer.hu

**Abstract:** Biocompatible ceramics are extremely important in bioengineering, and very useful in many biomedical or orthopedic applications because of their positive interactions with human tissues. There have been enormous efforts to develop bioceramic particles that cost-effectively meet high standards of quality. Among the numerous bioceramics, calcium phosphates are the most suitable since the main inorganic compound in human bones is hydroxyapatite, a specific phase of the calcium phosphates (CaPs). The CaPs can be applied as bone substitutes, types of cement, drug carriers, implants, or coatings. In addition, bioresorbable bioceramics have great potential in tissue engineering in their use as a scaffold that can advance the healing process of bones during the normal tissue repair process. On the other hand, the main disadvantages of bioceramics are their brittleness and poor mechanical properties. The newest advancement in CaPs doping with active biomolecules such as Mg, Zn, Sr, and others. Another set of similarly important materials in bioengineering are biopolymers. These include natural polymers such as collagen, cellulose acetate, gelatin, chitosan, and synthetic polymers, for example, polyvinyl pyrrolidone (PVP), polyvinyl alcohol (PVA), and polycaprolactone (PCL). Various types of polymer have unique properties that make them useful in different fields. The combination of CaP particles with different biopolymers gives rise to new opportunities for application, since their properties can be changed and adjusted to the given requirements. This review offers an insight into the most up-to-date advancements in the preparation and evaluation of different calcium phosphate–biopolymer composites, highlighting their application possibilities, which largely depend on the chemical and physical characteristics of CaPs and the applied polymer materials. Overall, these composites can be considered advanced materials in many important biomedical fields, with potential to improve the quality of healthcare and to assist in providing better outcomes as scaffolds in bone healing or in the integration of implants in orthopedic surgeries.

**Keywords:** biopolymers; calcium phosphate; composites; hydroxyapatite

## 1. Introduction

Calcium phosphate-loaded biopolymer composites are among most intensively and increasingly studied research areas, since they can be applied in a wide variety of forms by adjusting their physical–chemical properties to many requirements [1]. The appropriate mixture of biopolymer and bioceramic particles can provide a chemical composition that is similar to native bone and the extracellular matrix, containing inorganic minerals and organic collagenous material [2]. However, weak interfacial bonds have been reported to exist between biopolymers and bioceramic particles, which hinders the perfect formation of bioceramic–biopolymer composite scaffolds for bone repair implementations. In this context, applying certain bonding agents containing two different functional groups can build a so-called molecular bridge between the interface of biopolymer and bioceramic particles that could provide an ideal solution [3]. In this case, one of the functional groups

is organophilic in nature and can react with the polymers, while the other type of functional group can attach to the bioceramic surface to create a strong bond. Using these bonding agents, the mechanical properties of composite scaffolds could be improved.

Current general procedure for healing damaged bone structure may involve the use of bone grafts or substitutes [4]. Synthetic bone grafts or substitutes must have appropriate physical structure and mechanical as well as chemical properties. In applications involving high structural loads, It is also important that the bone substitutes can endure the biological conditions without the failure or degradation of implant materials. Fixation of orthopedic implants can lead to structural changes that may cause release of toxic or health-impairing particles or ions [5]. Bioceramics can be produced according to various methods which can generate different phases such as crystalline, polycrystalline, amorphous, or their composites [6]. The biological performance of CaPs is dependent on their physicochemical properties and determines their application potential [7]. Hydroxyapatite (HAp) is regarded as the most thermally and chemically stable phase among the CaPs; therefore, it is an ideal material to produce composites with polymers. Meanwhile, nanostructured CaPs are suitable for drug or gene delivery and carrier systems owing to their large surface areas [8]. It is also known that CaP scaffolds have hierarchical nano- and microstructures, which can provide outstanding assistance in bone healing. The integration of bioceramics into biopolymer matrices is able to mix the strength and osteoconductivity of calcium phosphate-based bioceramics with good mechanical characteristics and controlled bioresorbability of a polymeric matrix. The preparation of composites can be performed either by biopolymer infusion into the porous bioceramic scaffold [9] or by dispersing the ceramic particles into the base polymer solution [10]. In the latter case, the resulting dispersion can be further processed by electrospinning or spin coating onto the implants' surfaces to produce composite coatings, or the use of specific post-dispersion treatment to obtain the desired scaffold materials [11–14]. The main benefits of coating medical devices are the considerable enhancement of their biocompatibility and their long-term stability. In general, bioceramic particles or coatings are able to provide bioactive properties to the polymer matrix. The level of bioactivity can be adjusted by choosing the appropriate weight ratio, particle form and size, and the suitable dispersion of filler agents [15]. It has also been reported that a higher surface-area-to-volume ratio of ceramic particle fillers can cause higher bioactivity [16]. Addition of bioactive particles to bioresorbable polymers can also alter processes of polymer degradation, reportedly due to ion-exchange processes in biological environments [17,18]. This process reportedly imparts a pH-buffering effect at the polymer surface, thus altering the acidic polymer degradation. In similar research, it was observed that biopolymers with embedded HAp/CaP particles degraded consistently owing to water penetration at the interfacial areas [19]. Studies have described how the degradation and resorption mechanisms of composite scaffolds allow the cells to adhere, proliferate, and secrete their extracellular matrix (ECM), whilst the scaffolds gradually degrade, providing new space for bone cells or tissues to grow [20].

This review aims to help further the general understanding of the usefulness of CaP-loaded biopolymer composites and their possible applications as novel, high-performance biomaterials that can stimulate bone-healing mechanisms and implant integrations. In addition, we discuss the characteristics of different biopolymers, elaborating on their advantages and disadvantages as well as their current implementation in including tissue engineering and the biomedical field. We also elaborate on the biological performances of composites and compare the different preparation methods as described by the most recent literature.

## 2. CaP Containing Biopolymer Composites in Bone Tissue Engineering

CaP fillers can be prepared synthetically, using chemicals to precipitate the different phases of calcium phosphate. It is widely recognized that the precipitation parameters and the post-treatment of the resulting powder determine the final phase structure [21]. In most cases, CaP powder is a mixture of the different phases (monetite, brushite, hydroxyapatite,

tricalcium-phosphate, or even amorphous apatites) in various ratios. The different phases show different morphologies that affect their chemical and biological performances. The phase purity can be improved by applying appropriate post-treatments. The quality of calcium precursors used also affects the morphology, and the particle size and shape [22]. Other methods applied for CaP preparation include hydrothermal [23–25], sol-gel [26,27], electrochemical [28], solid phase powder milling [29–31], and spray freeze-drying techniques [32,33]. Additionally, because the human bone contains trace elements such as Mg, Zn, Sr, etc., the mineralization of CaP particles with such bioactive ions makes them more biocompatible. Generally, the ionic substitutions can be rendered cationic by replacing the $Ca^{2+}$ with $Mg^{2+}$, $Zn^{2+}$, $Sr^{2+}$, $Si^{4+}$ ions, or anionic when the $PO_4^{3-}$ groups are substituted by $CO_3^{2-}$ or fluoride anions. The preparations, thorough characterizations, and properties of these materials have been exhaustively studied and reported in numerous papers and summarized in reviews [34–40]. A further purpose of ionic substitution is to achieve antibacterial properties, using silver or other specific additives [41,42]. Furthermore, there is an emerging effort to prepare CaP particles from natural sources, such as fishbone [43,44], oyster shell [45], eggshell [46,47], mussel shell [48], snail shell [49], as well as bovine origins, and marble [50]. Reports on the naturally derived CaP powders indicate that these alternatives are more environmentally friendly, sustainable, and cost-effective compared with synthetic preparations. In addition, waste recycling is an urgent contemporary issue, since waste generation puts a huge burden on the environment. These natural preparation methods also provide good solutions in this context, able to turn unwanted waste into important functional material. Another advantage of using these materials is that trace elements such as Mg, Na, K, and Sr as well as carbonate anions can be found in CaPs prepared from organic source, which is important for bioactivity [51]. In this case, the preferred preparation method has been calcination, which can generate HAp powders with high crystallinity, in a process capable of fast and economic production requiring less chemical consumption [51]. However, it is important to mention their disadvantages, since the CaPs obtained from natural sources may contain contaminants harmful to the human body, which cannot be eliminated by heat treatment. Moreover, the concentrations and ratios of trace elements are strongly dependent on the source material, and living organisms contain different amounts of minerals depending on species, age, and other factors. These factors are hard to control, so the reproducibility of these CaPs is problematic. This fact may impede their applicability in clinical or pharmaceutical use, as these industries demand the strictest standards for biomaterials.

*2.1. Composites Prepared with Biopolymers from a Natural Source*

2.1.1. Collagen-Based Composites

Collagen is a protein-based biopolymer. It is well known that bone is a mixture of apatite and collagen. To mimic the structure of the bone, numerous studies have been conducted trying to reveal the best apatite–collagen composite constructions and preparation methods [52]. However, the characteristics of the apatite–collagen composites developed so far cannot come close to the properties of natural bone. It can be said that bone itself is a relatively active tissue with outstanding self-regenerative potential. Therefore, bone defects can be successfully repaired by filling the damaged area with bioresorbable ceramic composites that accelerate the bone tissue regeneration process. Collagen in its cross-linked form can be assembled into networks with highly organized 3D structures that are suitable as bone scaffolds. These 3D scaffolds reportedly can degrade enzymatically in the human body over time. Their degradation rate can be altered by modifying the cross-linking [53]. The mixing of collagen with bioceramics such as CaPs has been discussed as a method to increase their mechanical properties. Zhong et al. [54] investigated the potential osteo-immunomodulatory impacts of ion-substituted HAp. Zinc and strontium ions were incorporated into HAp using a collagen template, according to a biomimetic method. It was reported that the developed composite could generate a beneficial microenvironment by stimulating macrophages. Furthermore, the composite showed a promoting effect on

the osteogenic differentiation of bone marrow mesenchymal stem cells. According to the literature, collagen–HAp could be a suitable scaffold for bone graft, resembling natural bone and providing excellent bioactivity. However, it has been noted that type I collagen is highly biodegradable and has poor mechanical strength [55,56]. In other research work, it was described that the combination of HAp with CaO then mixed with collagen resulted in slower degradation when employed as a bone graft in post-operation bone regeneration [57]. In most cases, the HAp–coll composites were prepared via sol-gel [58,59], electrospinning [60], or spin coating [61]. However, these methods require cytotoxic organic solvents and the HAp particles tended to aggregate and were unevenly dispersed within the fibrous matrix of collagen, thus restricting their clinical applicability. Zou et al. [62] developed a greener method for preparing such composites, reporting that the resulting environmentally friendly product had 40 times higher mechanical strength than others previously reported, and possessed excellent microstructure similar to natural bone. They used phosphate-buffered saline (PBS) and ethanol solution instead of organic solvents. The co-electrospinning of collagen with HAp dispersion provided a composite with a homogeneous fibrous structure. Interestingly, the HAp nanoparticles were preferentially oriented along the longitudinal direction of the collagen fibers, imitating the nanostructure of bones. The biocompatibility of the prepared composite fibers was studied in vitro, using human myeloma cells (U2-OS). The examined HAp–collagen scaffolds exhibited very high porosity with interconnected and irregular porous networks, and contained low-crystalline HAp nanoparticles which were homogeneously incorporated into collagen fibers [63]. In vitro studies also proved that these scaffolds were biocompatible and ensured excellent cell viability in terms of cell adhesion and proliferation [52]. Another observation was that the increase in HAp concentration within the composite affected neither cell growth nor bone generation [52,58]. Moreover, the studied HAp–collagen composites resorbed more easily compared with the pure ceramic scaffolds, because they were reabsorbed by osteoclasts [63,64]. In vivo tests demonstrated the formation of new tissue formation, as well as scaffold degradation [63,64]. It was also discovered that the composite scaffolds prepared by electrospinning provided apatite-forming capacity, and the higher HAp amount led to better in vitro bioactivity owing to the fibrous structure [64]. The in vivo biological characterizations of the HAp–collagen composites induced fast bone healing. In a recent work, Itoh et al. [65] performed in vivo studies on beagle dogs. According to their report, the developed implants were able to induce bone regeneration and new bone formation.

### 2.1.2. Gelatin-Based Bioceramic Composites

Gelatin is a protein-based, natural biopolymer that id easily commercially accessible. It has two different types, A and B [4]. Originally, gelatin is irreversibly hydrolyzed from collagen, and it is widely used as a gelling aid in the pharmaceutical, food, and cosmetics industries. The major advantages of gelatin are its good biocompatibility, promotion of cell adhesion, and ease of modification with different functional groups by coupling with cross-linkers or other ligands. Thus, gelatin-based composites are ideal candidates in tissue engineering [66]. In a very recent work, Bartmanski et al. [67] developed a novel biocompatible injectable composite that can be applied either as a bone-to-implant bonding material or as a bone graft. They used a composite containing hydroxyapatite, gelatin, and two types of transglutaminases as a cross-linking agent. All samples had satisfactory mechanical strength close to that of natural bones. Elsewhere, Thiyagarajan et al. [68] reported the thermal degradation mechanisms and kinetics of gelatin–HAp composites. They found that the composite affected the bone-healing process in a manner dependent on the amorphous calcium phosphate microstructure as well as the deposition kinetics. Incorporation of bioactive metal ions in HAp forthe production of gelatin–HAp composites resulted in better biological performance. It is important to understand the degradation mechanisms of gelatin–Hap composites, so they can be tailored to meet specific requirements. The researchers also discovered that addition of niobium ion into the HAp structure led to a faster degradation rate in gelatin–HAp composites than in the undoped HAp

sample. Hossan et al. [69] prepared and studied HAp–gelatin composites in a different way, by crystallographic and morphological characterizations that confirmed the generation of a micro-porous hydroxyapatite–gelatin composite scaffold. The pores within the scaffolds were interconnected, and the HAp particles also exhibited micro-porous morphology, which provided the elongated interfaces that are prerequisite for physiological and biological responses and integration into the adjacent tissues. According to their FTIR spectra, the crosslinked composite incorporated chemical bonds between the gelatin and the HAp particles. The TG/DTA data revealed that the gelatin–HAp composite was very stable, and the degradation temperature of gelatin in the composite was nearly 300 °C.

### 2.1.3. Chitosan-Based Bioceramic Composites

Chitosan is formed by chitin deacetylation. It is a biocompatible, biodegradable, and non-toxic polymer that possesses numerous useful biological characteristics, and is a useful polysaccharide in the biomedical field [70,71] thanks to its adequate resistance in biological environments and its solubility in many organic acid solvents. However, its drawbacks include is the fact that it exerts any biological effect only in an acidic medium due to its poor solubility at neutral or basic pH. Moreover, the scaffold must have optimal microstructure to support fast cell growth [72]. In general, chitosan can be formed into porous structures that can be used in cell transplantations or tissue regeneration. The porous structure of chitosan provides enough space for bone cells to grow and to induce new bone generation. As a composite, it can be used in combination with different CaP phases to treat bone defects in tissue engineering. The most frequently applied CaP phases in these composites are hydroxyapatite (HAp), α- and β-tricalcium phosphates, (TCP), and the nanocrystalline or amorphous apatites (see Figure 1). The combination of natural chitosan with pure or modified calcium phosphates has great potential in many biomedical fields including orthopedic, dental, and drug-carrier applications [73]. In these systems, the chitosan network acts as a matrix and encloses the CaP particles [74]. the addition of CaP into a chitosan polymer matrix improved the biodegradability, osteoconductivity, and mechanical strength of the composite [75,76]. Thus, it represents an ideal approach to improve the performance of these composite scaffolds [77–80]. Many research works can be found in the scientific literature regarding the preparation and development of chitosan–HAp composites, and have consistently proven the beneficial effect of chitosan on the enhancement of the properties of CaP nanoparticles [81,82]. Kong et al. [83] produced a novel highly porous nano-HAp–chitosan composite scaffold, and reported that this new composite possessed better biocompatibility than pure chitosan. The mechanical characteristics of these composites were also better because of the increased interactions between chitosan and the nano-HAp particles. Moreover, they exhibited excellent osteoconductive properties, and the porous composite underwent almost complete biodegradation. Kjalarsdóttir et al. [82] reported the bone-modeling capacity of CaP-added chitosan composites. They examined the bone-healing characteristics of injectable composite samples in vivo, using Sprague–Dawley rats. Their results indicated that the bone-regeneration effect increased with mechanical stimulation of the bone tissue using chitosan- or chitooligosaccharide-containing implants, and the rate of bone healing was dependent on the degree of deacetylation of the chitosan. However, another observation was that the chitosan–CaP soft composites as void fillers in bone were not perfect candidates for short-term osteogenesis stimulation, but they were usable as a container of chitosan and accelerated osteogenesis in the vicinity of the implant site. Radwan et al. [84] prepared chitosan–CaP scaffolds to prevent post-operative osteomyelitis. The CaP particles were prepared in situ within the chitosan matrix, and the composites assessed both in vitro and in vivo. The experimental results revealed that the composites allowed sufficient drug release rate over three days, and promoted osteoblast cell differentiation and proliferation. Meanwhile, the composites' antibacterial effects were shown by decreased bacterial count, and they reduced the inflammation in bone tissues. In other work, Osmond et al. [85] developed a tunable chitosan–CaP composite as a cell-instructive dental pulp capping agent. According to their very thorough investigations, the prepared

composites had appropriate compressive modulus, biocompatibility, and odontogenic capacity for application as a regenerative dental composite in the future. Örlygsson et al. [86] in their current research chitosan–CaP-based composites investigated their mineralization effect in critical-sized bone gaps in sheep models. These injectable composites contained two types of commercially available bone cements in given concentrations, and their performances were compared in sheep tibia, monitoring new bone formation, calcification, and their effect on surrounding tissues. The authors concluded that these composites could induce sustained bone formation and be replaced by newly formed bone tissue, according to the long-term study. In a very interesting study, Torres et al. [87] recently reported on the successful development of a chitosan network incorporating a high quantity of CaP particles that were usable as printable inks. The novel method for preparation was so-called robocasting (a low-temperature additive manufacturing technique). They also aimed to toughen these composites with silk fibroins, which reportedly improved the scaffolds' mechanical strength, and showed that silk fibroins embedded into the chitosan matrix promoted the metabolic activity of osteoblast cells. Obviously, for biomedical or clinical applications it is crucial that their mechanical properties meet the required strict standards. Taking this into account, the toughening of these types of composites is crucial and many research groups have focused on this area, with good and promising achievements [87–89]. The strength of composites is also dependent on their pore size and distributions For example, Iqbal et al. [90] produced dicalcium phosphate dihydrate (DCPD) composite scaffolds with different DCPD concentrations (0–50 wt%), and reported that the increasing DCPD content resulted in higher crystallinity and reduction in pore size and distribution, observing mainly entwined and closed porosity.

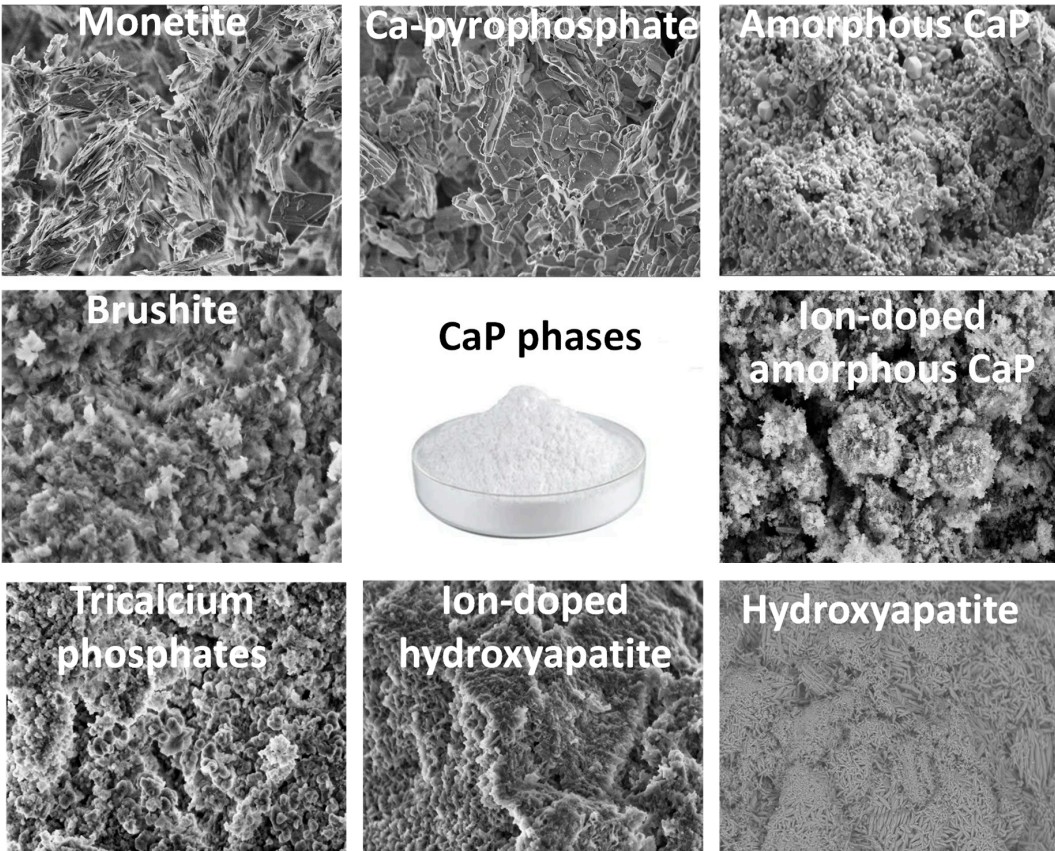

**Figure 1.** Illustration of different CaP phases that can be applied as filler materials in polymeric composites, such as Monetite, Brushite, Ca-pyrophosphate, Tricalcium phosphates (TCP), Hydroxyapatite (HAp), Amorphous calcium phosphate (ACP) as well as biomineralized/ion-doped ACP and HAp [11,21,37–39].

It is noteworthy that another type of preparation method in which the chitosan was infiltrated into the porous HAp scaffold by blending resulted in deterioration in the mechanical properties of the composite scaffold, owing to the weakened interfacial bond between chitosan and the HAp matrix. Theoretically, the molecular weight of chitosan also affects its mechanical characteristics. The chitosan scaffolds of high molecular weight had higher compression strengths than the polymer of medium molecular weight. Additionally, in vitro biocompatibility studies proved that the CaPs containing chitosan scaffolds had remarkably increased osteoblast cell growth on the scaffolds [91]. The cytotoxicity measurements revealed that all the composite scaffolds had perfect cytocompatibility behavior with better cell attachment and a higher proliferation rate of osteoblast cells [58,60,61]. The highly porous composite scaffolds (with pore size between 100 and 200 μm) allowed the 3D arrangement of cells that were able to diffuse into the composite [77]. It is worth mentioning that chitosan–CaP composites can be applied not only as scaffolds but also as coatings [92,93]. Zanca et al. [92] deposited chitosan–collagen–CaP composite layers onto AISI 304 type stainless steel using a galvanostatic method and characterized their morphological, structural, degradative, and ion-releasing properties and their biological performances. They found that the coating provided good corrosion protection and the viability tests on MC3T3-E1 cells confirmed their excellent biocompatibility. In another recent work, Zarif et al. [93] combined two types of coating preparation methods, namely radio-frequency magnetron sputtering (RFMS) and matrix-assisted pulsed laser evaporation (MAPLE). In their research, the CaP layer was deposited by RFMS by changing the temperature of the substrate, following the chitosan deposition by MAPLE onto the previously deposited CaP coating. According to their report, the substrate's temperature during deposition greatly affected the incorporation of the chitosan polymer, while the degree of chitosan infiltration had a significant influence on the chemical and physical characteristics of composite coatings as well as on their adherence strength to the substrate. The adherence became weaker after the chitosan deposition.

### 2.1.4. Alginate-Based Bioceramic Composites

Alginate is a natural polysaccharide and is extensively applied in bone tissue engineering, for example as the previously discussed chitin and chitosan. Alginate is non-toxic, non-allergenic, biocompatible, and biodegradable, and has a good scaffold-forming ability. A main advantage of alginate is that it can easily be converted into many forms such as hydrogels, microspheres, microcapsules, sponges, foams, or fibers. This feature can boost the applications of alginate in many areas, including tissue engineering and drug delivery. Sun et al. [94] developed and discussed various preparative methods, indicating that alterations of alginate structures can be useful for adjusting their biological and mechanical properties for many prospective applications. These physical and chemical alterations can be made by incorporating other molecules, such as growth factors and peptides [95,96]. Lin et al. [97] described an alginate matrix with HAp filler that functioned as a good porous scaffold material. They developed the composite using a phase-separation technique that improved osteosarcoma cell adhesion. The incorporation of HAp particles into the alginate matrix noticeably improved its mechanical strength, as well as inducing cell adhesion and proliferation on the porous scaffolds. In vivo studies have also been performed on alginate-HAp composites used as fillers in bone defects. New bone generation was perceived in the case of an alginate–HAp composite scaffold, as reportedly the alginate molecules formed anionic complexes that were able to adsorb important factors discerned by integrins from osteoblasts [96–98]. Bjørnøy et al. [99] similarly reported that alginate–calcium phosphate (CaP) composites could be used as potential scaffold materials in bone. Mineralization of alginate with dicalcium phosphate dihydrate (DCPD) or brushite was performed by crystal seeds that controlled the formation and supersaturation of the mineral. It was reported that the minerals within the composite material transformed from brushite to hydroxyapatite during immersion in simulated body fluid, implying its bioactive properties. Sancillio et al. [100] prepared HAp-strengthened alginate polymer

biocomposites for use as bone fillers. In their work, HAp particles were embedded into the alginate solution containing a gelling agent for the controlled release of calcium ions, to demonstrate the mineralization and differentiation capacity of human dental pulp stem cells seeded onto the scaffolds. They concluded in their results that the investigated cells expressed osteogenic differentiation-related markers and promoted calcium deposition and biomineralization on the scaffolds. You et al. [101] in their recent work studied calcified cartilage regeneration with homogeneous hydroxyapatite–alginate composite hydrogels. They tested the theory that HAp could trigger chondrocytes to secrete the characteristic substance of calcified cartilage. To test the biological properties of the prepared composites, chondrocyte viability and proliferation, extracellular matrix production, and mineralization capacity were measured with and without HAp particles. The alginate–HAp composite hydrogel scaffolds with highly porous characteristics were prepared by 3D printing. In conclusion, the obtained results confirmed the theory that the HAp particles embedded into alginate hydrogel could stimulate chondrocytes to secrete calcified substances in vitro and in vivo, and also revealed that these composites can be used in 3D bioprinting and for osteochondral regeneration. In another work, Sikkema et al. [102] summarized in a very detailed way the various uses and possible preparation methods of alginate composites. They reported that these composites are widely utilized as biomedical scaffolds in bone tissue engineering, for medical devices, in drug delivery, for wound dressing, and as protein-resistant coatings. Moreover, alginate and its composites can be used for surface modification of other types of biomaterials. Abundant discussion has described many ways of preparing alginate–CaP composites as a scaffold material, and its use creating innovative and important structures in tissue engineering [103–105], and as drug carriers [106]. The properties of scaffolds were found to be dependent on the content and ratio of alginate and CaPs, while their microstructure and the preparation methods were also important factors. The density and the porosity of these scaffolds were mainly determined by the amount of alginate. The diffusion of alginate into the porous scaffold network caused pore closure, resulting in improved mechanical properties. It was also revealed that alginate–HAp scaffolds accelerated bone healing, did not cause inflammation, nor had any carcinogenic effect.

Considering all the relevant literature data on this topic, it has been reported that alginate–CaP composites can also be prepared as coatings or films, using techniques such as dip coating, spin coating, electrophoresis or electrodeposition. The addition of HAp into dip-coated alginate layers enhanced their stability and biocompatibility [107] and their mechanical properties, showing an antibacterial effect as well as reduced capacity for water permeability [108]. Applying electrodeposition for preparation of alginate–CaP composites, it was discovered that the alginate molecules attached to HAp particles and promoted their dispersion. Alginate had layer-forming and binding abilities that enabled the formation of a well-adhered and homogeneous composite [109]. In other research work, the electrophoretic deposition of this composite was employed to modify the 3D porous Ti6Al4V scaffolds [110] or to prepare coating on titanium substrates [111]. In this method, the thickness of the coating could be controlled by the applied voltage and deposition time and was dependent on the deposition solutions applied. The microstructure of the coating was also dependent on the amount of HAp in the layer; higher HAp content caused agglomeration or cluster formations [111].

### 2.1.5. Cellulose-Based Bioceramic Composites

Cellulose is the most widely available natural carbohydrate-based polymer. It is biocompatible, biodegradable, hydrophilic, and insoluble in water. It can be found in all plants, herbs, trees, and in cell walls. Furthermore, it has sufficient mechanical strength that makes it an ideal raw material in a wide range of applications, such as clothing, paper, biofuel, and biomedical fields [112–114]. Its derivative, cellulose acetate (CA) is obtained by the acetylation of cellulose. The generated acetyl groups change the biostability of the original polymer; nevertheless, cellulose acetate can also be regarded as a biodegradable

polymer. It is non-irritant, heat-resistant, nontoxic, and relatively less hygroscopic in nature. CA can be partially or completely acetylated [115]. The combination of CA and calcium phosphate-based ceramics has given rise to new developments in biomedical applications, since the properties of the composite materials can be altered. The CaP-added cellulose-based composites can fuse the main features of the cellulose and calcium–phosphate minerals, endowing new features to the composite in a synergistic way [116]. In recent work, Tabaght et al. [117] synthesized by a newly developed dissolution and precipitation technique a biocompatible HAp–cellulose composite that can be used as a bone substitute. Generally, the co-precipitation technique is an important way to prepare HAp–cellulose composites. Sivasankari et al. [118] described a chemical precipitation method to produce HAp embedded in CA–polyetherimide composites. They proposed an easy and economical method to produce such materials that can be used either as adsorption membranes or biomedical applications. The chemically precipitated HAp particles were thoroughly dispersed into a cellulose acetate–polyetherimide polymer mixture by a phase-inversion method. According to the thorough characterization, it was discovered that the embedded HAp nanoparticles improved the thermal and hydrophilic features of the composite. The biocompatibility was tested on THP-1 human monocytic leukemia cells, which provided promising results. Nicoara et al. [119] used co-precipitation combined with ultrasound exposure to prepare HAp–bacterial cellulose–Ag composites both in situ and ex situ. They measured outstanding bioactivity as well as the antibacterial effect of the samples, and reported that the composites had a porous and homogenous structure with excellent water-absorption capacity. Chen et al. [120] developed biomimetic HAp–cellulose nanocomposites with good mechanical properties and investigated their mineralization ability. They discussed the mineralization process of HAp, followed by the biological secretion of nanocellulose by *Acetobacter xylinum*. The experiments revealed that the newly developed cellulose molecules significantly promoted the nucleation rate and provided a uniform distribution of HAp particles.

CaP–cellulose scaffolds can be prepared by many methods, which determine the main characteristics of the materials. Thus, these composites can be utilized in various ways and forms. For example, Palaveniene et al. [121] hydrothermally synthesized an osteoconductive, 3D HAp–cellulose composite scaffold with 85% porosity that may be ideal for healing bone tissue. Using a microwave-assisted hydrothermal method, Pieper et al. [122] developed a HAp–cellulose biofilm that had excellent thermal stability. Gao et al. [123] produced deacetylated 3D porous cellulose–HAp–polydopamine microsphere coating scaffolds with proper adherence. These novel scaffold materials are reportedly able to induce osteogenic cell differentiation by in vitro mineralization [124]. In addition, Abdelraof et al. [125] recently proposed a green synthesis method to prepare bioactive HAp–cellulose composites. They used bacterial-derived cellulose (BC) and eggshell to produce the HAp. In vitro tests in SBF solution proved that all composites induced bone-like apatite formation, and the cell viability tests showed their proper biocompatibility. In another work, Elsayed et al. [126] prepared fibrous scaffolds from cellulose acetate (CA) via electrospinning incorporating modified hydroxyapatite and Cu ions in various concentrations. The purpose of the developed material was to accelerate and improve the healing rate of wounds. Its antibacterial behavior was tested against Gram+ and Gram- bacteria. The largest inhibition zones that were measured corresponded with the highest Cu content. In addition, the viability tests on human fibroblast cell lines demonstrated a high proliferation rate. Hence, this nanofibrous scaffold containing ion-modified ceramic composite can be an ideal candidate for wound dressing. Sofi et al. [127] prepared HAp–Ag–CA composite in a novel way, as another type of antibacterial scaffold. They developed acetate-free nanofibers by alkaline deacetylation of previously prepared electrospun fibers, and the nanofibers were also combined with hydroxyapatite and silver nanoparticles. The deacetylation of composite nanofibers resulted in spontaneous hydrophilicity. The novelty of this work is that they overcame the difficulty of cellulose electrospinning into nanofibers, since the post-modification of its acetate-derived fibers in an alkaline solution is an easier process. According to the cell

viability tests, the samples were biocompatible. The antimicrobial activity of the nanofibers was also tested and results showed that the developed composites could restrain bacterial proliferation. All the results proved the great efficiency of these nanofibers in soft and hard tissue engineering, with an additional antibacterial effect. In another very recent work, Athukorala et al. [128] synthesized bacterial cellulose–hydroxyapatite nanocomposite hydrogel as an effective 3D cell-culture matrix and evaluated its biological properties. The structure of composites closely resembled the native extracellular matrices (ECM), proving it to be a perfect substrate for cell culture. The biocompatibility tests indicated a high rate of cell proliferation. In general, as proven by numerous scientific papers, CaP-added cellulose composites can be widely applied in processes of bone or soft tissue engineering, and represent a rapidly growing research area [129,130]. In a study, Bayir and coworkers [131] developed a novel, highly porous bacterial cellulose (BC)–HAp composite structure, which combined the good mechanical characteristics of cellulose and the bioactive features of HAp. They found that the composites containing the smallest HAp particles provided the best biocompatibility and a good swelling ratio, thus making these composites suitable for 3D cell culture. In conclusion, they stated that these BC–HAp composites could have possible applications in drug delivery, regenerative medicine, and cell therapy. More recently, Shi et al. [132] developed a new method by modifying bacterial cellulose using Ca-gluconate as a carbon source during the bacterial synthesis of BC. This method provided more nucleation sites for advanced mineralization in simulated body fluid. The spherical HAp particles entirely filled the porous 3D BC network structure. With this method, the mechanical strength and biocompatibility of the composite were significantly improved and the preparation process was simplified, compared with the conventional method. Figure 2 illustrates the commonly used natural polymers that can be mixed with CaP particles in different concentrations.

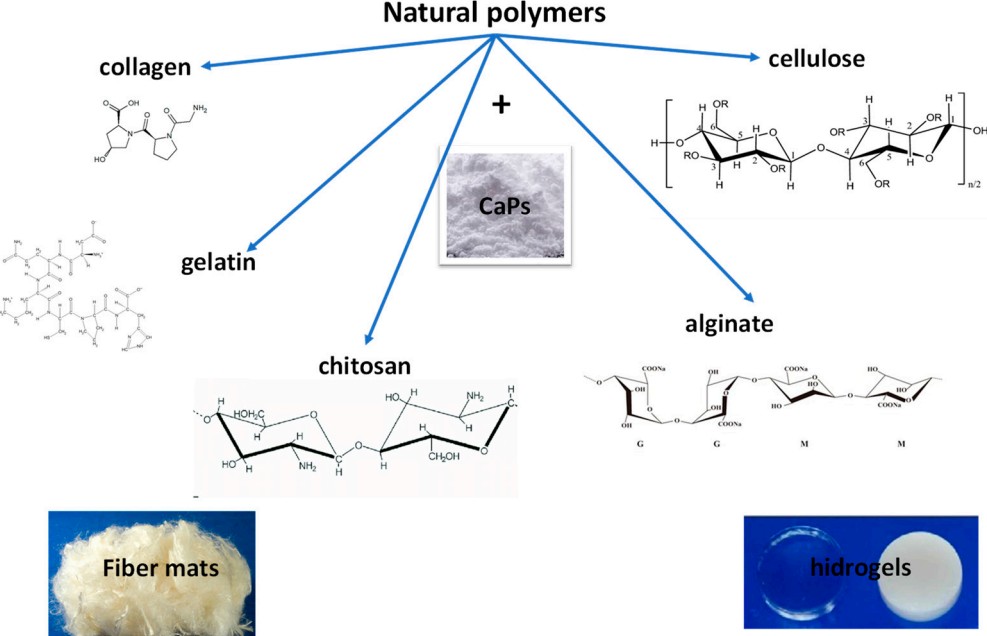

**Figure 2.** Graphical illustration of the most common natural polymers that can be used as CaP-containing composites. The applications of these materials are extremely broad, since they can be produced either as hydrogels or fiber mats.

In Table 1, we summarize the reported properties, forms as well as common application areas of CaP-loaded natural polymer composites.

**Table 1.** CaP containing natural polymer composites in bone tissue engineering.

| Polymer Matrices | Properties | Form | Applications |
|---|---|---|---|
| Collagen | Biodegradable, biocompatible, cytocompatible, bioactive, low mechanical properties | Hydrogels, 3D scaffolds, film coatings | Bone grafts, bone and tissue engineering, pharmaceutical |
| Gelatin | Biodegradable, biocompatible, non-immunogenic, bioactive, injectable, low mechanical properties | Hydrogels, 3D scaffolds, film coatings | Bone-to-implant bonding, bone grafts, tissue engineering, cartilage, pharmaceutical |
| Chitosan | Biodegradable, biocompatible, water-soluble, slightly antibacterial and antioxidant, low mechanical properties | Hydrogels 3D scaffolds, membranes, film coatings | Bone and tissue engineering, energy and environmental applications, food packaging, pharmaceutical, drug carrier |
| Alginate | Biocompatible, water-soluble, high viscosity, suspending agent, film-forming ability | Hydrogels 3D scaffolds, membranes, film coatings | Bone and tissue engineering, prosthesis, dental molds, and impression materials, pharmaceutical, drug carrier |
| Cellulose | Biocompatible, bioactive, biodegradable, non-water-soluble, high water permeability, film-forming ability, mechanical strength, osteoconductivity | Hydrogels 3D scaffolds, membranes, film coatings | Bone and tissue engineering, drug delivery, bone graft, drug carrier |

## 3. Composites Prepared with Synthetic Biopolymers

### 3.1. Polylactic Acid (PLA) Composites (Biomass-Based)

Polylactic acid is a synthetic biopolymer that is very versatile. It can be prepared from abundant renewable materials, is biodegradable, and exhibits thermoplastic behavior. In principle, PLA comprises lactic acid monomers which contribute to its polymeric structure. PLA is a very promising material in many healthcare applications, for example in bone tissue engineering, regenerative medicine, dental materials, drug carriers, orthopedic implants, cancer therapy, skin care, and tendon healing, as well as in medical devices. Importantly, it is a 3D printable polymer. In addition, the degradation by-products of PLA are harmless to either humans or the environment. In tissue engineering, PLA has a major role in fulfilling many strict requirements. It can promote hard tissue regeneration in bone-grafting processes. The most up-to-date research works are focusing on the integration of tissue-engineered or synthetic bones with natural bone. The synthesized materials reportedly promote osteogenesis and angiogenesis with adjacent tissues [133]. PLA combined with hydroxyapatite or other calcium phosphate phases (CaPs) is a very promising material in this area, since the CaPs can prompt osteogenesis by activating the osteoblasts or pre-osteoblastic cells [134]. The CaP–PLA composites can unite the unique properties of each component, and the components within the composite will influence and change the physical and mechanical characteristics of one another. Hatano et al. [135] for example, developed a PLA–HAp composite and studied its mechanical properties. This

composite was prepared with a cost-effective, user-friendly hot pressing technique. The results revealed that the composite with 80 wt% HAp content had a value of elastic modulus similar to that in human bones. However, increasing the HAp content above 80% resulted in inferior structural features. One possible reason for this phenomenon may be that the HAp crystals tend to aggregate or accumulate. On the other hand, it was also reported that PLA embedded with HAp of high molecular weight boosted osteoblast growth and cell viability. Russias et al. [136] reported the change in the physical characteristics of PLA caused by HAp incorporation. They produced a composite via a high-velocity stream using micro-sized HAp particles. This technique resulted in homogenous distribution of the HAp along the PLA surface. According to the measurement data, the surface roughness in the PLA–HAp composites increased five times compared with pure PLA. In addition, the hydrophilicity also improved. These alterations are very important in terms of the enhancement of protein adsorption and interconnection with the extracellular matrix. The enhanced wettability led to better hydrophilicity, which is advantageous for cell attachment. Moreover, the rough surface provides a better environment for pre-osteoblast adhesion, proliferation, and differentiation [137].

In more recent work, Salamanca et al. [138] fabricated polylactic acid/β-tricalcium phosphate composites by three-dimensional printing and used fused deposition modeling (FDM) to create the fibers, then investigated their osteoblastic-like cell performance. The addition of β-TCP to PLA changed the mechanical characteristics of the material, and the tests revealed that the tensile and elongation strengths decreased, the hardness did not change significantly, and cell proliferation was higher. Alksne et al. [139] compared the biological performances of 3D-printed polylactic acid–HAp as well as polylactic acid–bioglass (BG) composite scaffolds and concluded that the composite that included bioglass had better properties than the HAp–PLA composite, thus the PLA–BG composite scaffolds may be a better choice for preparation of synthetic bone tissue.

It has been extensively reported that CaPs can effectively be used as artificial bone substitutes and are often combined with other materials to solve problems related to the weak mechanical characteristics of most CaPs. In addition to tissue engineering, applying bioceramic–biopolymer composites as coatings (with different thicknesses) on implant materials is another huge and extremely important part of biomedical applications. In these composites, the CaP properties (phase, morphology, particle size, and shape) and the incorporation method into the hybrid material both determine the biocompatibility of the final structure. Birgani et al. [140] comparatively investigated a monolithic composite comprising nano-sized CaP–PLA and a CaP-coated PLA for their effectiveness in promoting the proliferation and osteogenic differentiation of bone cells. The composite was prepared by physical mixing and extrusion. Both types of materials were bioactive and supported cell proliferation. This study proved the importance of CaP content in osteogenic differentiation, and that the role of the incorporation method into the hybrid material was less notable. Nevado et al. [141] prepared PLA–biphasic calcium phosphate (BCP) composite filaments with a diameter of 1.7 mm by using hot-melt extrusion with a single-screw extruder. The BCP particles were prepared according to the solution–combustion method, and it was confirmed that the obtained composite had high porosity. The in vitro tests on the fibers confirmed the formation of apatite phases on their surface, and they were also biocompatible. Sahu et al. [142] synthesized biodegradable PLA from lactic acid monomers by ring-opening polymerization (ROP). Calcium phosphate and magnesium phosphate nanoparticles were embedded into a PLA polymer to examine their mechanical and rheological characteristics. According to the results of the mechanical tests, the developed nanocomposites might be a perfect applicant for bone implants, since the tensile strength of both types of nanocomposites was close to that of human bones. Pérez et al. [143] evaluated the mechanical properties of polylactic acid (PLA)-based composites containing different calcium phosphates. The investigated CaP phases were hydroxyapatite (HA) or β-tricalcium phosphate (β-TCP). The performed characterizations showed proper filler dispersion for composites obtained by the extrusion method. In more recent work pub-

lished by this research group [144], the mechanical characteristics of a PLA–HAp composite were investigated. In this case, the PLA matrix was incorporated with hydroxyapatite in different concentrations and prepared as film or fiber. The developed fibers had higher tensile strength than the films, and faster degradation owing to the thinner cross-section. Akindoyo et al. [145] investigated the in vitro biocompatibility of modified composites prepared from PLA and HAp. The aim of surface modification was to enhance the dispersion of HAp in the polymer. The performed cell viability tests showed a positive effect of HAp surface modification on cell adhesion and proliferation. The addition of HAp offered better cell attachment and proliferation in the PLA matrix. In addition, the modification of HAp caused no cytotoxic effect on the PLA–HAp composite. Pandele et al. [146] also worked on the preparation of PLA–micro-structured HAp composite layers. The composites were produced from a polymeric solution in which hydroxyapatite was evenly dispersed with different content by ultrasonication and solvent evaporation. They observed a noticeable decrease in the crystallinity of the composite films compared to the pure polymer. The hydroxyapatite crystals had no significant effect on the degradation temperature of the composite film. However, in other research work, Yudyanto et al. [147] focused on the preparation of nano HAp–PLA composites from natural materials using a sonication technique. The results similarly confirmed the composites to be biocompatible. The sample with a ratio of 90 wt% of nano HAp and 10 wt% of PLA showed the highest bioactivity and biodegradability. Interestingly, in a more recent work, Carvalho et al. [148] developed a new type of PLA–calcium phosphate-based biocomposite with magnetic properties for tissue engineering. This work presents an alternative method to increase the magnetic sensibility of PLA-based biocomposites. They incorporated iron-doped biphasic calcium phosphate powders into the PLA matrix and compared the properties of composites containing different amounts of powders or pure PLA. The composites with iron-doped CaPs exhibited typical ferromagnetic features and even with low concentrations of filling particles had better tensile strength than pure PLA. Moreover, these composites showed good cytocompatibility. Considering their preparation methods, the PLA–CaP composites can also be prepared by spin coating [149] or electrospinning techniques [150–153].

### 3.2. Polyvinylpyrrolidone (PVP) Composites (Petroleum-Based)

Polyvinylpyrrolidone (PVP) is another easily-prepared type of non-toxic synthetic polymer with good biocompatibility [154]. In addition, it has a very similar structure to that of proteins, which is an important factor in the context of dental implants and bone substitutes [155,156]. Even so, only a few studies on the applicability of PVP-based composites are available, mainly because of the composites' rapid solubility in water-based solutions. In light of this fact, their applications in bone tissue repair and engineering are restrained. However, there have been published studies where PVP was used as a filling agent or template to aid precipitation of CaPs and change their morphology or grain size and shape. Dau et al. [157] examined the in vivo characteristics and levels of integration as well as the degradation of a ready-to-use bone graft for possible surgical use. They concluded that the bone substitute incorporating cross-linked PVP-based hydrogel had delayed degradation. Nathanael et al. [158] also used a PVP-assisted method to obtain hydroxyapatite nanorods with adjustable aspect ratio and bioactivity. They managed to prepare highly crystalline and adequately uniform hydroxyapatite nanorods via a hydrothermal method, generating nanorods with a high aspect ratio (length–width). The aspect ratio of the nanorods was higher in the presence of PVP and increased with its increasing content. Cell viability studies in vitro showed very promising results for nanorods with a high length–width ratio. In an interesting work by Mukhanova et al. [159], they investigated the impact of the molecular weight of polyvinylpyrrolidone on the structure and morphology of materials based on substituted hydroxyapatite used for bone implants. The results revealed that the PVP polymer agent affected the crystal size and the thermodynamic stability of the formed structure, and higher molecular weight reduced the grain size. Another investigated composite type was biphasic calcium phosphate (BCP)–polyvinylpyrrolidone

(PVP)–graphene oxide (GO), as a promising material for biomedical implants [160]. In this case, a potential advantage of the 2D graphene oxide (GO) was its promotion of cell differentiation while enhancing the mechanical strength of the scaffolds. They discovered that with increasing GO content, the mechanical properties of scaffolds also improved. The BCP nanoparticles were homogeneously distributed onto the graphene oxide surface. The biocompatibility tests revealed no adverse effect, which proved the good biocompatibility of the composite scaffold. Guesmi et al. [161] synthesized CaP and CaP–PVP composites by low-temperature precipitation. Recognizing that the application of bioactive composites for bone repair is an important issue that needs to be addressed, the authors proposed a method of grafting the surface of CaP microcrystals with PVP by wet precipitation to produce new types of composite scaffolds suitable for bone tissue engineering. They found that the prepared composites could be used in the repair of bone fractures. The addition of PVP caused a decrease in the CaP's crystallinity, and its thermal characteristics also changed, implying strong interaction between the components. Homogenous aggregations of CaP particles within the polymeric matrix were observed and the surface roughness increased. Bioactivity tests on the composites also revealed promising results. In an up-to-date study reporting the preparation of hydroxyapatite incorporated into PVP–aloe vera composite, Mathina et al. [162] addressed a very current issue regarding waste recycling. They used crab shells as waste materials to prepare the HAp, and the developed composite had seemingly enhanced mechanical and antibacterial characteristics, and at the same time increased biocompatibility. The incorporation of HAp into the polymer composite increased its mechanical strength, and the addition of aloe vera further improved the antibacterial effect and biocompatibility. The antibacterial efficiency of the composite was tested against Gram+ and Gram- bacteria, and its biocompatibility was assessed on MG 63 cells. The researchers concluded that the enhancement of mechanical characteristics of the PVP, as well as the antibacterial and biocompatible properties of the aloe vera in the composite, may make it useful as a potential therapeutic material with many biomedical applications.

### 3.3. Polycaprolactone (PCL) Composites (Petroleum-Based)

PCL is another very important and versatile synthetic and biodegradable polymer. It can be prepared from renewable sources by the chemical treatment of saccharides. The degradation process of PCL involves the hydrolysis of its ester linkages in a physiological environment. and as such it has great potential as an implantable biomaterial. It may also be an ideal candidate for medium- or long-term implantable materials. Another positive feature of this polymer is that a wide choice of drugs can be incorporated within PCL beads, and controlled release in drug carrier and delivery carrier applications can be achieved [163]. The investigation of the combination of PCL with bioceramic particles such as CaPs is a related topic of current research, and intensive efforts have been made to elaborate the most ideal composite for biomedical applications. In tissue engineering, PCL–CaP composites can be used as scaffolds [164–169] as well as coatings [37,170–178].

Juan et al. [164] investigated bone cell formation on polycaprolactone–bioceramic 3D porous scaffolds, and their bioactivity. They used an enhanced solvent-casting–particulate-leaching method to obtain 3D scaffolds with high porosity. The experimental data revealed that the CaP (either HAp or TCP) powder increased the hydrophilic characteristics of the scaffolds. The degradation of the scaffolds was also accelerated by addition of CaP. Moreover, the scaffolds also demonstrated excellent in vitro biocompatibility. Xu et al. [165] fabricated 3D artificial polycaprolactone (PCL)–HAp bones by the fused deposition modeling technique. Bauer et al. [166] recently prepared PCL-coated multi-substituted calcium phosphate bone scaffolds with improved properties. They used ionic doping within the hydroxyapatite phase to imitate the chemical composition of the bone mineral. Sr-doped as well as Mg- and Sr-co-doped CaP scaffolds with different strontium and magnesium content were obtained by the hydrothermal method. The PCL-coated CaP scaffolds were prepared by vacuum impregnation. The thorough biocompatibility tests carried out revealed that these composites were highly bioactive, and uncovered the positive effect of $Sr^{2+}$

ions on the differentiation of the investigated cells, in accordance with the histology results. Fazeli et al. [167] prepared PCL scaffolds coated with nanobioceramics using a 3D printer, to promote osteogenic cell differentiation. They reported that the prepared PCL scaffolds had relatively low bioactivity and poor cell attachment on their surfaces. However, using an easy post-modification method with hydroxyapatite and bioglass (BG), improved cell proliferation and attachment could be achieved since the bioceramic coating was able to provide a more suitable surface for cell adhesion and growth. In addition, they can induce faster osteoconduction and osteointegration than pure PCL. The PCL–HAp–BG scaffold showed the highest cell viability and capacity for bone formation, which can be attributed to the synergistic effect of HAp and the bioglass. They stated that this tri-component 3D-printed scaffold had promising prospects in bone tissue engineering applications. Another research work [168] discussed the preparation and characterization of a novel calcium phosphate (CaP)–polycaprolactone (PCL) scaffold with graded composition and high porosity. The scaffold contained a dense HAp–β-TCP inner layer, a porous HAp–β-TCP transition layer, and a porous PCL–(HAp–β-TCP) outer layer. These multi-layered ceramics were made by gel-casting, and the outer composite layer was prepared by solvent casting and particle leaching. Ressler et al. [169] in a very recent work reported the preparation of PCL–silicon-doped multi-phase calcium phosphate scaffolds, in which the CaP component was derived from cuttlefish bone. The preparation involved a simple and inexpensive hydrothermal method, and the prepared scaffold was coated with PCL. During the procedure, the very porous structure of cuttlefish bone was maintained within the composite scaffold. The relatively high compressive strength of the obtained composite enabled its use as scaffold for non-load-bearing applications. Moreover, the cytocompatibility assessment of the composite scaffold revealed its non-cytotoxic properties.

The use of PCL-based ceramic composites as coatings on load-bearing implant materials is a rarely discussed topic in the scientific literature. Chunyan et al. [170] developed a PCL–HAp composite coating by mixing hydrothermal and dipping methods. The purpose of the coating was to prevent the degradation of substrate Mg alloy AZ31, a bioresorbable implant material. The composite coating consisted of nanorod-shaped HAp crystals and PCL that infiltrated into the space between the HAp crystals. Compared to the pure HAp coating, the adherence between the PCL–HAp composite coating and the substrate Mg alloy was significantly stronger It is noteworthy that the corrosion rate of the HAp-coated sample slowed ten-fold after being infiltrated by PCL. The results revealed the PCL–HAp composite coating could ensure a more successful barrier for the Mg substrate compared with the pure HAp coating. Moreover, the PCL–HAp composite coating in their study demonstrated better biocompatibility that was more suitable for cell adhesion than the pure HAp coating. However, from the perspective of clinical application, PCL–HAp composite did not show significant antibacterial properties. There are other works aiming at improving the corrosion properties of biodegradable metallic magnesium alloys by depositing PCL–CaP composites onto their surfaces [171–173]. The composite coating can be deposited onto other metallic implants, such as titanium alloy, to make them more biocompatible or even bioactive [37,174]. The deposition can be carried out in different ways, such as by dip coating [175,176], in situ sol-gel process [174], spin coating [37,177], or electrospinning [37,175]. Montanez et al. [177] prepared an advanced biocomposite coating by mixing PCL with layers of different CaP phases (hydroxyapatite, brushite, and monetite—derived from a biomineral called otolith), and also multiwalled carbon nanotubes in different concentrations. The biocomposite coating was deposited onto Ti6Al4V by spin coating. Results revealed that the increase of the carbon nanotube content caused a change in the microstructure of the CaPs, leading to the formation of brushite, monetite, and hydroxyapatite, and slightly improving the adherence of the coating to the substrate. Another interesting application might be involve the use of PCL–CaP composites as drug-release coatings on implant surfaces. Bose et al. [178] aimed to explore the effect of PCL coating on the release kinetics of the drug alendronate in vitro. According to their theory, the PCL coating could minimize the immediate release of alendronate from plasma-sprayed

Mg-doped hydroxyapatite (HAp)-coated commercially pure titanium. They used the PCL coating to modulate the release kinetics, and reported that the application of a PCL coating could control the release kinetics of alendronate from the HAp-coated titanium implants. This discovery can positively affect countless patients worldwide who have damaged bones due to osteoporosis. Another published work on the use of composite coatings as drug-delivery systems was produced by Iynoon Jariya et al. [179], who developed an advanced drug-delivery structure with vaterite microsphere, graphite oxide (GO), and reduced GO (rGO) incorporation in the PCL matrix as a layer on $TiO_2$ nanotube-coated Ti. The different composite coatings were deposited by dip coating. The vaterite–rGO/PCL composite coating exhibited a low dissolution rate and had sufficient bioactivity in physiological conditions. All composite coatings promoted cell viability, growth, and proliferation. The vaterite–rGO/PCL composite coating could ensure a slow and steady release of drugs with adequate bioactivity and biocompatibility at the implant surface, which makes it a promising candidate for coatings in bone tissue implants. In Figure 3, we present the most commonly used synthetic polymers that can be combined with CaP particles with different content ratios, as well as some composite preparation methods.

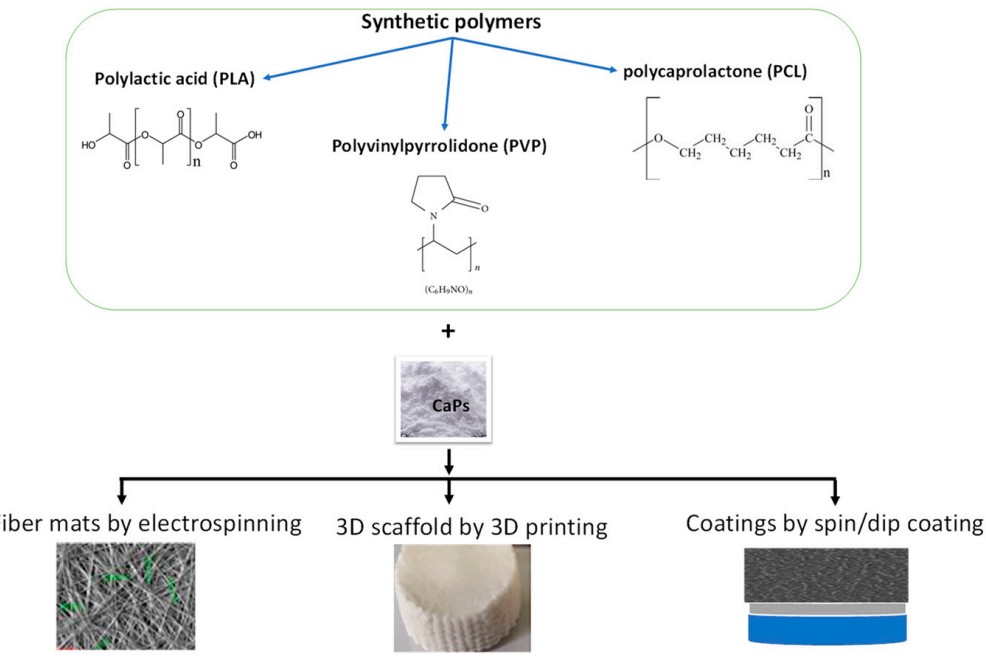

**Figure 3.** Graphical illustration of the most frequently used synthetic polymers that can be used as CaP-containing composites, their possible application forms, and preparation methods.

Table 2 demonstrates the reported properties, forms as well as common application areas of CaP-loaded synthetic polymer composites.

**Table 2.** CaP-containing synthetic polymer composites in bone tissue engineering.

| Polymer Matrices | Properties | Form | Applications |
|---|---|---|---|
| Polylactic acid (PLA) | Biodegradable, biocompatible, good mechanical properties, non-water soluble, hydrophilic, slow biodegradation rate | 3D-printed scaffolds, films, fibers. | Bone grafts, bone and tissue engineering, pharmaceutical, regenerative medicine, drug carriers, dental material, coatings on orthopedic implants |

**Table 2.** *Cont.*

| Polymer Matrices | Properties | Form | Applications |
|---|---|---|---|
| Polyvinylpyrrolidone (PVP) | Biodegradable, biocompatible, highly water soluble, low mechanical properties | hydrogel, fibers | Surfactant, filling agent in tissue engineering, bone graft, cartilage, pharmaceutical |
| Polycaprolactone (PCL) | Biodegradable, biocompatible, non-water soluble, hydrophobic, good mechanical properties, implantable | 3D-printed scaffolds, membranes, films, fibers | Bone and tissue engineering, drug carriers, drug delivery, dental material, coatings on orthopedic implants, pharmaceutical |

## 4. Blended Polymer Composites

Theoretically, a polymer blend combines two or more polymers to form a new material with different physical properties. The physical, chemical, and mechanical characteristics as well as the biological performances of a certain polymer can be changed or even improved by combining them with other polymers with different features. These various types of blends thus have significant roles in different biomedical fields. A combination of the most beneficial characteristics of each polymer can provide a new hybrid blend, with significantly better properties that might differ from the properties of each single component [180].

### 4.1. Synthetic-Natural Blended Polymer Composites

Natural and synthetic polymer blends can be regarded as a new class of materials, with enhanced mechanical characteristics and biological properties compared with one-component materials [180–183].

PVP-based composites represent one type of synthetic–natural polymer blend. Since the PVP polymer itself has poor mechanical strength as a hydrogel, its applications are restricted. Because of this, its combination with other polymers is very frequent [184–187]. Fadeeva et al. [188] described the development of composite films comprising bioresorbable polymer blends of PVP and sodium alginate (SA) with HAp filler. The PVP–SA–HAp composite films were crosslinked and exhibited swelling characteristic of hydrogels. Interestingly, it was observed that the PVP–SA–HAp hydrogel film composite with in situ synthesized HAp fillers was cytotoxic, which can be explained by the presence of reaction by-products of mainly ammonia and ammonium nitrate. In contrast, cell viability was increased in the coatings with ex situ synthesized HAp filler. As a conclusion, the authors claimed that the developed hydrogel film composites (PVP–SA–HAp ex-situ) could be applied as medical membranes or as wound dressings. In other research, Kandasami et al. [189] used a combination of hydrothermal and electrospinning methods to prepare zinc and manganese-doped HAp–CMC–PVP composites suitable for use in bone repair. They confirmed the successful generation of fibers and the inclusion of Zn and Mn-doped HAp in the fiber structures. These composites showed good physical and mechanical characteristics, and were also biocompatible with good hemocompatibility.

PCL-based blends represent another common group of synthetic–natural polymer blends. They are regarded as the most favorable composites in bone tissue engineering (BTE) because of their interconnected porosity and good mechanical, chemical, and biological properties [190]. For example, Wang et al. [191] prepared PCL–HAp–collagen 3D scaffolds and evaluated their biological performance. They found that the developed scaffolds had outstanding osteoinduction ability and high rates of cell proliferation able to induce fast bone regeneration. Linh et al. [192] investigated PCL–gelatin (GE) polymer blend fibers prepared by electrospinning, and their composite with HAp. In addition, they studied the in vivo bone generation capacity of composite in rats, and observed increased bone formation. Supported by the results of the animal tests they firmly claimed that

these scaffolds are suitable for use as membranes in tissue engineering and even in bone tissue engineering. In a recent work, Kichi et al. [193] developed PCL–gelatin-forsterite nanocomposite coatings on titanium substrate by electrospinning. Based on the cell viability experiments they revealed that the bioactivity of composites increased with the decreasing gelatin content. Ebrahimi et al. [194] recently reported the preparation of PCL–collagen–HAp composites as scaffolds. The biodegradable scaffolds were prepared by 3D printing combined with fused deposition modeling (FDM). The authors reported that although all the developed scaffolds facilitated cell proliferation and differentiation, the best results were achieved in the case of HAp- and collagen-containing scaffold coatings. This again confirms that the HAp and collagen could intensify the biological capacities in a synergistic way.

Meanwhile, significantly fewer publications exist on CaP-added PLA-natural polymer blends. One of the existing articles, by Cai et al. [195], describes production of HAp–chitosan–PLA nanocomposites by in situ precipitation. The HAp particles were homogeneously dispersed into the chitosan–PLA matrix. It was revealed that PLA addition into the chitosan polymer strongly affected the nucleation and the growth of HAp crystals. The team evaluated the mechanical properties of the scaffold and the results showed that the PLA addition made the composite more stress resistant, with better mechanical properties and higher elastic modulus. This can make the composite advantageous for surgical applications. In an earlier work, Liao et al. [196] made HAp–collagen–PLA composite bone scaffolds using a biomimetic method. They reported that the HAp and the collagen combined into mineralized fibrils and presented a 3D porous structure similar to the microstructure of cancellous bone. According to the characterization tests performed, the composite was bioactive. Finally, they drew the conclusion that this scaffold is a promising material for the clinical repair of large bone failures. Recently, Rahman et al. [197] developed high porosity hydroxyapatite–chitosan composite scaffold with gelatin PLA, by the sol-gel technique followed by lyophilization. The HAp content in the composite scaffold changed between 5 and 20 wt%. The pore size decreased with increasing HAp content. The antibacterial and biocompatibility tests confirmed that the scaffold resisted the investigated bacteria and was not cytotoxic. the mechanical tests revealed that the composite was weaker than human bone, but owing to its highly porous structure it can be applied as spongy bone graft or substitute. Another interesting research work [198] describes 3D printing of PLA–collagen–minocycline–hydroxyapatite scaffolds that had both antimicrobial and osteogenic effects. In this work, the PLA scaffolds were 3D-printed then surface functionalized with collagen, minocycline, and bioinspired citrate-hydroxyapatite nanoparticles. These novel scaffolds exhibited uniform microporous structure, good wettability, and proper compressive strength. The addition of minocycline provided an antibacterial effect with sufficient rate of antibiotic release to inhibit biofilm formation. Moreover, the HAp content promoted cell adhesion and proliferation.

### 4.2. Synthetic/Synthetic Blended Polymer Composites

The possibilities arising from blending two or more synthetic biopolymers are wide, since their combination can endow new and more promising properties to the composite. These polymer blends are usually hydrogels that are mainly used in tissue engineering. For example, the PVA hydrogel alone is not sufficient to serve as a biomaterial in either clinical or biomedical applications, since the attachment of cells onto PVA hydrogel is impeded by to its highly hydrophilic nature [199]. Some researchers have concentrated on blending PVA with other biopolymers, such as poly(vinylpyrrolidone) (PVP) [200] to create composite hydrogels with enhanced biological performance. The miscibility and mechanical properties of PVA–PVP hydrogel blends have already been exhaustively examined [201,202]. Moreover, PVP–PVA interactions have been discussed in numerous publications, as the constructed polymer blend combines the properties of both polymers resulting in new, interesting, and innovative features [203,204]. It has also been reported that the strong hydrogen bond between the carbonyl group of the pyrrolidone ring in

PVP and the hydroxyl group in PVA is responsible for the good solubility of both PVP and PVA in water, and for their perfect miscibility and blending at any ratio. PVP-PVA blends are often used in medicine as skin-dressing components and in electrochemistry as membranes [205]. It should be mentioned that the major disadvantage of PVA is their instability in physiological environments. PVA–PVP blends, however, might subdue this restriction since the strong interchain hydrogen bond enhances their stability, so these hydrogel blends could become very stable in biological media. PVA and PVP both possess outstanding biocompatibility and biodegradability [206,207].

As a perfect example, Ma et al. [208] developed PVA–PVP–HAp composite hydrogels by repeated freezing and thawing, and evaluated the effects of HAp content on the physical, chemical, and biological characteristics of the composite hydrogels with and without HAp particles. They showed that the HAp-containing composite hydrogels had denser network structures, lower water content, larger storage modulus, and higher dehydration activation energy compared to the pure PVA–PVP blend. It has also been reported that PVA–PVP–CaP composites can be used as cartilage replacements [209–211].

PCL-based composites represent another type of synthetic polymer blend. PCL can also be blended with PLA polymer. In a very recent work, Ismail et al. [48] examined the combination of a PCL–PLA polymer blend with HAp derived from green mussel shells. The preparation of composites was performed by the chemical blending method. They studied the effects of HAp content on the mechanical strength and degradation rate of the polymer blends, and found that increasing the HAp and PLA content in the matrix caused improvement in the mechanical characteristics of the developed composites. In addition, the degradation rate of the biocomposite polymer blend also increased. In other research work, Åkerlund et al. [212] also reported that the combination of PLA, PCL, and HAp produced adjustable, biocompatible, and biodegradable composite fibers using extrusion for fused filament fabrication (FFF) printing. In this case, the PLA and PCL acted as supporting polymers and the HAp filler was incorporated to improve the biological properties. All composites had higher mechanical strength than human bone, and the HAp filler increased the polymer's degradation rate significantly, which could be beneficial for faster healing when support is required for a shorter period.

These types of polymer blends can also be applied as coatings on metallic implants. Recently, Etminanfar et al. [213] deposited HAp–PEG-*b*-PCL bilayer composite coatings onto NiTi alloy. The HAp layer was electrodeposited, and the PEG-*b*-PCL polymer blend was drop-cast. The study showed that the electrodeposited HAp coating had a dense inner layer and a porous outer layer, and the polymer blend could infiltrate into the porosities of the HAp. The cell viability tests confirmed that the composites were bioactive. The drug-delivery capacity of the coating was investigated by loading ibuprofen onto the composite film and then following its release profile. In another work, Mystiridou et al. [214] developed an innovative composite bone scaffold composed of a PLA–PCL polymer blend, and both HAp and barium titanate were used as fillers. The composite filaments were produced by a single-screw melt extruder, and the 3D composite scaffolds were prepared using the fused deposition modeling (FDM) technique.

## 5. Future Perspectives and Possible Limitations

Bioceramic–biopolymer composites are extremely important in the biomedical field, and can be used in a wide variety of forms. Therefore, there is an intense and constantly growing demand to develop high-quality, bioactive and biodegradable composites. The urgent need to improve the quality of biomaterials for clinical use has stimulated researchers to develop scaffolds, bone grafts, and implant coatings with better chemical and biological properties and enhanced mechanical stability. Summarization of the present state of discoveries in this area has revealed that preparation methods, the different types of polymers and polymer blends, and the morphological structure and content of the CaP filler all had significant influence on the final properties of the obtained scaffolds, hydrogels, or coatings. The addition of bioactive ions into the base CaP phases is a promising way to

make the powder more biocompatible and more easily accepted by the body, however, the correct mineral concentrations and ratios are important, and it is hard to find the optimum compositions.

According to the existing literature, these composites are often applied as bone-scaffold materials and for bone grafts in bone tissue engineering, while only a few studies have covered their applications as coatings on implant materials in orthopedic surgery or any medical devices. The thorough literature survey also highlighted that there remain critical problems that need to be addressed before these materials can safely be used in clinical applications. The main limitations of the conventional scaffold-preparation techniques include poor reproducibility, uneven pore sizes, different pore shapes, restricted interconnectivity between pores, difficulty obtaining the required geometries and forms for the scaffolds, the need to use toxic organic solvents, and the difficulty eliminating the residual contaminants from the scaffold. However, these problems can be solved by 3D-printing preparation, which represents the most innovative method to produce porous, sponge-like structures [36,103], since it can provide a precisely tailored structure and is able to control pore size and shape as well as the surfaces of polymer matrices. It is also crucial to choose appropriate raw materials and to optimize the applied printing parameters and concentrations of polymer and filler. The exact interaction mechanisms and the possible synergistic effects of different polymer blends remained to be explored.

The crucial factors for scaffold materials are their toughness and porosity [215]. It is also important to study and determine how the daily activities of humans affect the fatigue life of porous scaffold implants [216]. In order to meet the necessary standards, the mechanical strength and the structure of the material should perfectly reflect the construction of natural, spongy bones.

The exact mechanism of bone regeneration and new bone formation at the interface of bone graft or implant coating and natural bone needs to be explored and understood in greater detail. Appropriate porosity is important for better integration and improved osteoconductivity. It is necessary to determine the optimum porosity rate, since higher porosity facilitates bone ingrowth and causes deterioration in mechanical strength.

Overall, biodegradable polymers or polymer blends are ideal scaffold matrices, since they promote cell adhesion and differentiation, and accelerate bone tissue repair. However, the exact mechanisms of their degradation, the excretion rate of their metabolites, and their effects on the human body remain unclear. For CaP-added and/or drug-loaded polymer composites, there are insufficient available data to follow the release rate and subsequent dissolution of the filler materials. Moreover, it is challenging to adjust the appropriate concentrations of the additives to obtain optimal effects. In other words, the published evaluations of biological performances of the different porous scaffold composites remain insufficient, and more in vivo studies and mechanical characteristics testing are required so that these materials can be safely adapted into human bodies.

CaP-loaded biopolymer composites face different requirements, restrictions, and limitations when applied as coatings. Commonly used deposition methods have been found effective in vitro. However, similarly to scaffold composites, they are not yet used in clinical practice, despite a few of them being commercially applicable. The ideal implant coatings must fulfill all the important conditions, including structural integrity, appropriate surface roughness, mechanical and chemical stability, sufficient adhesion, porosity, biocompatibility, bioactivity, and improved osseointegration features. Meanwhile, the biocomposite coating must be reproducible and should degrade gradually when the implant has been osseointegrated and the bone healing is complete. However, the degradation mechanism of coatings on metallic implants in human body conditions can be completely different than widely investigated simulated body fluids.

It is well known that implant-related infections and implant failures continue to occur, with considerable negative effects on both and the healthcare system. In addition, the exact mechanism and rate of the degradation processes are unexplained to date, because of the lack sufficient in vivo examinations and clinical trials. This lack of information continues

to delay their clinical applicability. Another similarly important factor is the interaction and adhesion strength between the substrate material and the composite coatings. The adherence of coatings to substrates remains insufficient and needs to be improved. One possible solution could be to improve the adherence of the coating by inserting an intermediate bonding layer between the coating and the substrate. Overall, the main limitations worth mentioning are the low reproducibility, low stability, insufficient tribological and mechanical properties, and the importance of the selecting the most suitable raw materials, the optimum process, and ideal parameters to achieve the best coatings. In some cases, toxic organic solvents must be used, and the residual contaminants are hard to remove. These drawbacks need to be addressed in the future by the development of more advanced functional scaffolds, and coatings with extended lifetimes and better stability in biological environments or human bodies.

As another future perspective regarding clinical applications of different ceramics, it is worth mentioning that interesting advanced ceramic couples are being developed to minimize or avoid postoperative failures in total hip prosthesis. These novel materials are ceramic-on-ceramic couplings such as $ZrO_2$-on-$ZrO_2$, $Al_2O_3$-on-$Al_2O_3$, and $Si_3N_4$-on-$Si_3N_4$. The biomechanical properties of these couplings were tested by measuring Tresca stress, which can help in the selection of the best candidate for clinical applications. [217]

## 6. Conclusions

In this review, we have summarized the most recent advances in the field of calcium phosphate-containing biopolymers, focusing on the different polymer types, the possible preparation methods, and their main characteristics and biological performances. According to the thorough search of the literature on the different biopolymer–calcium phosphate composites, it is apparent that this is an extremely important area from biomedical and clinical points of view. Although an enormous number of publications exist in the scientific literature, and the research in this area is very intensive and exponentially growing, it is obvious that there is still a requirement to develop new types of composites with improved mechanical, chemical, and biological performances. In tissue engineering, these materials can be applied as scaffolds, hydrogels, and drug carriers. in bone tissue as synthetic grafts or fillers. In orthopedic applications, these materials are ideal candidates as coatings on the surfaces of implant materials and even on medical devices.

Considering all the obtained data regarding the characteristics and applications of the discussed CaP-added biopolymer composites, we can draw the conclusion that these materials must be further investigated so that they can be safely adopted in human bodies and produced on an industrial scale. To obtain this goal, it is first essential to understand the degradation rate and mechanisms of these composites, which can allow remodeling of the bone tissue and help us avoid future post-operative intervention. The information summarized in this review can highlight a pathway to developing novel and innovative composites that can meet high standards of quality in the clinical and pharmaceutical industries.

**Author Contributions:** Conceptualization, M.F.; writing—original draft preparation, M.F.; writing—review and editing, M.F.; supervision, C.B. and K.B; methodology and investigation, M.F., K.B., and C.B. All authors have read and agreed to the published version of the manuscript.

**Funding:** National Research, Development and Innovation Office–NKFIH OTKA-PD 131934.

**Institutional Review Board Statement:** Not applicable.

**Informed Consent Statement:** Not applicable.

**Data Availability Statement:** Not applicable.

**Conflicts of Interest:** The authors declare no conflict of interest.

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
