# Peer review of "Calcium Phosphate Loaded Biopolymer Composites—A Comprehensive Review on the Most Recent Progress and Promising Trends"

_coatings, doi:10.3390/coatings13020360_

Round 1

Reviewer 1 Report

The review paper entitled “Bioceramic loaded biopolymer composites for biomedical applications-a comparative review on most recent progresses” is in an interesting field that is rapidly progressing. Therefore, reviewer papers on this topic are highly required. Below authors can find comments and suggestions that can help authors to improve the quality of the review. I hope the authors will find them useful.

The abstract is very well written.

The review paper is nicely constructed and a wide variety of biopolymers is covered.

As the authors mentioned in the abstract “The newest advancement in the CaPs when they are doped with active biomolecules such as Mg, Zn, Sr and so on.” More articles that used ion-substituted CaP in polymer matrices should be included in the review. The influence of these ions on the biological performance of the scaffolds could be mentioned and discussed.

In addition, CaP used in composite scaffolds that were obtained from biogenic sources can be included in the review as they contain trace elements and are considered multi-substituted CaPs.

The “CaP source” in image 1 is misleading. It could be changed to “CaP type” or something similar.

The future perspectives and conclusions should be expanded. It can be expanded to the ion-substituted CaP used in polymer matrixes as this is the biomimetic approach that might be a pathway to follow.

References are checked and up-to-date references have been considered. However, ~42% of the references are published in the last 5 years. This percentage should be increased and more references published in the last 5 years should be considered.

After the authors consider the above given suggestion, the manuscript should be of high interest for publication.

Author Response

We would like to thank the thorough revision of our manuscript and for all the constructive comments.

The review paper entitled “Bioceramic loaded biopolymer composites for biomedical applications-a comparative review on most recent progresses” is in an interesting field that is rapidly progressing. Therefore, reviewer papers on this topic are highly required. Below, authors can find comments and suggestions that can help authors improve the review's quality. I hope the authors will find them useful.

The abstract is very well written.

Answer:

We are grateful for the positive comment.

The review paper is nicely constructed and a wide variety of biopolymers is covered.

As the authors mentioned in the abstract “The newest advancement in the CaPs when they are doped with active biomolecules such as Mg, Zn, Sr and so on.” More articles that used ion-substituted CaP in polymer matrices should be included in the review. The influence of these ions on the biological performance of the scaffolds could be mentioned and discussed.

Answer:

Thanks for the constructive comments, we have thoroughly searched all the available scientific literature and we included and mentioned all the relevant papers on ionic substituted CaP-biopolymer composites that we found. We also described their biological performance that has been reported in the cited references.

In addition, CaP used in composite scaffolds that were obtained from biogenic sources can be included in the review as they contain trace elements and are considered multi-substituted CaPs.

Answer:

We have provided the requested additional information on CaPs from organic source as well as we mentioned the ion doped CaPs as well.

The “CaP source” in image 1 is misleading. It could be changed to “CaP type” or something similar.

Answer

We corrected it ti CaP phases.

The future perspectives and conclusions should be expanded. It can be expanded to the ion-substituted CaP used in polymer matrixes as this is the biomimetic approach that might be a pathway to follow.

References are checked and up-to-date references have been considered. However, ~42% of the references are published in the last 5 years. This percentage should be increased and more references published in the last 5 years should be considered.

Answer:

We have supplemented the future perspectives and conclusions to the best of our knowledge.

Reviewer 2 Report

1.      As your abstract's final sentence, include a "take-home" message.

2.      Keywords should have been reorganized alphabetically.

3.      Nothing truly unique in its current state. Because of the lack of novel, the current review looks to be a replication or modified literature. The authors must describe their novel in detail. This work should be rejected owing to a major issue.

4.      It is essential to summarize previous review' merits, novelties, and limitations in the introductory part to emphasize the gaps in the research that the latest review seeks to address.

5.      Related to ceramics application in biomedical application, especially in medical implant has been studied as computational simulation to faster results and lower cost advantages. The introduction and/or discussion part of an article should contain this crucial topic, according to the authors. In addition, to reinforce this explanation, the MDPI-recommended reference should be cited as follows: Minimizing Risk of Failure from Ceramic-on-Ceramic Total Hip Prosthesis by Selecting Ceramic Materials Based on Tresca Stress. Sustainability 2022, 14, 13413. https://doi.org/10.3390/su142013413

6.      The authors need to improve the discussion in the present article become more comprehensive. The present form was insufficient.

7.      Please include the limitation of the present review, it is missing. Include it before conclusion section.

8.      Further prospect and conclusion in line 739-760 recommended to be spitted.

9.      In the conclusion, please explain the further research.

10.   The authors should give additional references from the five-years back. MDPI reference is strongly recommended.

11.   Throughout the whole manuscript, the authors sometimes wrote paragraphs with just one or two phrases, which made the explanation difficult to understand. To make their explanation a full paragraph, the writers should expand it. It is advised to use at least three sentences in a paragraph, with the primary sentence coming first and the supporting sentences coming after.

12.   Due to grammatical mistakes and English style, English has to be proofread.

13.   After revision, provide a graphical abstract for submission.

Author Response

We are very grateful for all the constructive comments and for the very thorough review of our paper.

  1. As your abstract's final sentence, include a "take-home" message.

Answer:

We inserted the essence of the present review in the Abstract, as advised.

  1. Keywords should have been reorganized alphabetically.

Answer:

The keywords are re-organized.

  1. Nothing truly unique in its current state. Because of the lack of novel, the current review looks to be a replication or modified literature. The authors must describe their novel in detail. This work should be rejected owing to a major issue.

Answer:

We understand your opinion, however, to compile our review paper we previously made a very thorough research in all existing scientific databases (MDPI, Science Direct, Scopus, Springer, IOPscience..) and we found more than 500 papers that are closely related to calcium phosphate added different biopolymers in five years back. Around 10% of them are review paper. Since this area is extremely wide, the number of papers is huge. Browsing through the most relevant articles and reviews, our paper present new and useful information compared to the published ones. For example, there is no paper where the Authors systematically describes the most recent achievements in the preparation and characterization of most common biopolymers (especially concentrating on the synthetic and natural polymers and even their blends) incorporated with different CaP phases (pure and biomineralized alike). We also described their reported biological performances and applicability.

  1. It is essential to summarize previous review' merits, novelties, and limitations in the introductory part to emphasize the gaps in the research that the latest review seeks to address.

Answer:

We have cited the most relevant review papers throughout the whole manuscript, in the sections where they most fit.

  1. Related to ceramics application in biomedical application, especially in medical implant has been studied as computational simulation to faster results and lower cost advantages. The introduction and/or discussion part of an article should contain this crucial topic, according to the authors. In addition, to reinforce this explanation, the MDPI-recommended reference should be cited as follows: Minimizing Risk of Failure from Ceramic-on-Ceramic Total Hip Prosthesis by Selecting Ceramic Materials Based on Tresca Stress. Sustainability 2022, 14, 13413. https://doi.org/10.3390/su142013413

Anwser:

We have included the suggested reference in the future perspectives section.

  1. The authors need to improve the discussion in the present article become more comprehensive. The present form was insufficient.

Anwser:

We have included and cited more relevant and up-to date papers to improve and supplement the discussions. We hope, in this current form it will be sufficient and more informative.

  1. Please include the limitation of the present review, it is missing. Include it before conclusion section.

Anwser: We have discussed the limitations of these materials in the Section 5.

  1. Further prospect and conclusion in line 739-760 recommended to be spitted.

Answer: We have divided the section in two as suggested.

  1. In the conclusion, please explain the further research.

Answer: We provided some information on the future perspectives/researches need to be done.

  1. The authors should give additional references from the five-years back. MDPI reference is strongly recommended.

Answer: The addition of more recent references have been made.

  1. Throughout the whole manuscript, the authors sometimes wrote paragraphs with just one or two phrases, which made the explanation difficult to understand. To make their explanation a full paragraph, the writers should expand it. It is advised to use at least three sentences in a paragraph, with the primary sentence coming first and the supporting sentences coming after.

Answer: Thank you for the constructive comment, we have tried to re-phrase and expand the discussion in more detail as suggested.

  1. Due to grammatical mistakes and English style, English has to be proofread.

Answer: We have proofread our manuscript by grammar checkers and have it read by a native English speaker.

  1. After revision, provide a graphical abstract for submission.

Answer: The required graphical abstract is provided (Figure 4)

Reviewer 3 Report

It was evaluated the article “Bioceramic loaded biopolymer composites for biomedical applications-a comparative review on most recent progresses”. It is a simple review with a confusing title.

The topic is interesting and the info used is correct. But there is a scientific bias in the methodology, besides many concerns. This article seems a thesis that was compacted and transformed in an article. I recommend to work on it to improve the quality and reproducibility of the study.

- Please, explain what is a comparative review
- In general, the text is long with huge paragraphs

ABSTRACT: “This short review offers an insight into the most current” line 24. It is not a short review (around 25 pages) - also included in line 84.
- I considered the abstract incomplete.

- Where is the methodology used to elaborate this review?

- If it is a review, where the authors found the pictures 1-3 from?

- “In general, these composites should be further investigated regarding their safe clinical applicability. Moreover, the biomaterials’ degradation rate and mechanisms are very important issues because they can help us avoid future post-operations and allow remodeling the bone tissue.” Lines 757-759
This conclusion above did not concluded anything.

Author Response

It was evaluated the article “Bioceramic loaded biopolymer composites for biomedical applications-a comparative review on most recent progresses”. It is a simple review with a confusing title.

Answer: Thank you for the constructive comment, we have re-phrases and simplified the title.

The topic is interesting and the info used is correct. But there is a scientific bias in the methodology, besides many concerns. This article seems a thesis that was compacted and transformed in an article. I recommend to work on it to improve the quality and reproducibility of the study.

Answer: Thanks for your comment, we have tried to make some amendments in the manuscript, however, it’s original structure remained. The area of bioceramic-biopolymer composites is extremely wide, and it is hard to cover all the aspects and address all emerging issues. Here, we only focused on the possible preparation methods and the reported chemical, mechanical and biological performances of calcium phosphate loaded natural and synthetic and blended biopolymers. We hope you will deem our paper publishable in this corrected form.

- Please, explain what is a comparative review.

Answer: In this case we compared the characteristics of biopolymers, polymer blends incorporated with CaP particles. That is why the comparative adjective was used.

- In general, the text is long with huge paragraphs
ABSTRACT: “This short review offers an insight into the most current” line 24. It is not a short review (around 25 pages) - also included in line 84. - I considered the abstract incomplete.

Answer: We have made additions to the abstract and deleted the short adjective before review.

- Where is the methodology used to elaborate this review?

Answer: This is a good question. The methodology would be to show and categorize the different biopolymers and show how the CaP addition affect them according to the literature.

- If it is a review, where the authors found the pictures 1-3 from?

Answer: All the Figures are our own art. Tor chemical formulas we used free access and free-to-use chemical databases. In Figure 1 we presented our SEM images on the prepared CaPs by us.

- “In general, these composites should be further investigated regarding their safe clinical applicability. Moreover, the biomaterials’ degradation rate and mechanisms are very important issues because they can help us avoid future post-operations and allow remodeling the bone tissue.” Lines 757-759
 This conclusion above did not concluded anything.

Answer: We have re-phrased and supplemented the conclusion.

Round 2

Reviewer 1 Report

-

Author Response

We are grateful for the thorough and careful review of our manuscript and for the positive comments that helped to improve its quality.

Reviewer 2 Report

I have other comments as response to authors revised version specifically in yellow highlight as follows:

1.      Line 2-4, why the previous title is changed? Any specific reasons?

2.      Line 414 the authors mention about young modulus compared to nature cellulose. I think the authors should provide young modulus (and posion ration) comparison in form of chart or table for some biopolymer composite for make the content more enriched.

3.      In section 5, Future perspectives and possible limitations the authors giving additional explanation specifically to scaffold application. It is suitable for refer recent relevant study from Putra et al. as follow: Level of Activity Changes Increases the Fatigue Life of the Porous Magnesium Scaffold, as Observed in Dynamic Immersion Tests, over Time. Sustainability 2023, 15, 823. https://doi.org/10.3390/su15010823

Author Response

We would like to thank again for the grateful review and the constructive comments. We have made some changes accordingly and we hope that you will find this revised version acceptable for publication.

The answers are as follows:

I have other comments as response to authors revised version specifically in yellow highlight as follows:

  1. Line 2-4, why the previous title is changed? Any specific reasons?

Answer:

We have changed the title following the other reviewer’s advice because they found it a little “confusing”.  In this changed form, they deemed it better.

  1. Line 414 the authors mention about young modulus compared to nature cellulose. I think the authors should provide young modulus (and posion ration) comparison in form of chart or table for some biopolymer composite for make the content more enriched.

Answer:

Thank you for the constructive advice, and it would be a really great idea to summarize the values of Young’s modulus and also the Poisson ratios of composites. However, we think that the demonstration of a deeper mechanical characterizations of these composites could be a topic of another short review, because, in this case, we should collect and compare the values of all types of composites, not just the cellulose composites. Thus, this review would be so large and confusing, and hard to comprehend. So, instead, we have deleted the part that mentioned the Young's modulus, which was added to the paper after the first revision. In this paper we would like to talk about the mechanical properties in general way. We hope you will agree to this.

  1. In section 5, Future perspectives and possible limitations the authors giving additional explanation specifically to scaffold application. It is suitable for refer recent relevant study from Putra et al. as follow: Level of Activity Changes Increases the Fatigue Life of the Porous Magnesium Scaffold, as Observed in Dynamic Immersion Tests, over Time. Sustainability 2023, 15, 823. https://doi.org/10.3390/su15010823

Answer:

Many thanks for the suggestion, we have added and cited the reference in Section 5, as advised.

Reviewer 3 Report

It was re-evaluated the article “Calcium phosphate loaded biopolymer composites - a comprehensive review on the most recent progress and promising trends”. It is a simple review that had its titled modified (it is better).

The topic is interesting and the info used is correct.
The methodology is not reproducible.

Many concerns were raised:
- In general, the text continues long with extensive paragraphs
- line 86: “foster” - review the word

- Where is the methodology used to elaborate this review?

- The conclusion section is long and is generalized.

Author Response

We would like to thank again for the very thorough and careful review and the constructive comments and we also appriciate your concern about this paper. We hope that you will find this revised version acceptable for publication.

The answers to the comments are as follows:

It was re-evaluated the article “Calcium phosphate loaded biopolymer composites - a comprehensive review on the most recent progress and promising trends”. It is a simple review that had its titled modified (it is better).

Answer: Thank you, we did our best.

The topic is interesting and the info used is correct.
The methodology is not reproducible.

Many concerns were raised:
- In general, the text continues long with extensive paragraphs

Answer:

We understand your concern. We have tried to give a detailed and comprehensive review with as many citations as needed to cover the state-of-the-art in this particular topic. That required detailed descriptions and explanations which resulted in longer paragraphs. We think that a longer and more detailed description gives more useful information to the scientific community.

- line 86: “foster” - review the word

Answer:

Thank you for the observation, it was a wrong word choice, we changed it to “stimulate”

- Where is the methodology used to elaborate this review?

Answer:

To our knowledge, methodology is the study of research methods and also refers to the methods themselves used to achieve certain goals and the analysis of the procedures of examinations in a particular field. In this qualitative review, we made a systematic summarization with sufficient descriptive elements and relevant quotes from the most relevant scientific literature. Anyway, we have applied a similar methodology as the Authors used in many other review papers:

Levingstone, T.J.; Herbaj, S.; Dunne, N.J. Calcium Phosphate Nanoparticles for Therapeutic Applications in Bone Regeneration. Nanomaterials 2019, 9, 1570. https://doi.org/10.3390/nano9111570

Hou, X.; Zhang, L.; Zhou, Z.; Luo, X.;Wang, T.; Zhao, X.; Lu, B.; Chen, F.; Zheng, L. Calcium Phosphate-Based Biomaterials for Bone Repair. J. Funct. Biomater. 2022, 13, 187. https://doi.org/10.3390/jfb13040187

 Barinov, S.M. Calcium phosphate-based ceramic and composite materials for medicine. Russ Chem Rev 2010, 79(1) 13–29.

We think that our methodology is clearly introduced: "This review offers an insight into the most current advancement in preparation and evaluation of different calcium phosphate-biopolymer composites, highlighting their application possibilities which largely depend on the chemical and physical characteristics of both CaPs and applied polymer materials. We summarized the possible preparation methods and the reported chemical, mechanical and biological performances of calcium phosphate-loaded natural and synthetic and blended biopolymers, classifying them by polymer types.

Moreover, we also added to the end of the manuscript: Author contribution: methodology and investigation: M.F, K.B, C.B"

- The conclusion section is long and is generalized.

Answer:

We think that the Conclusions section describes adequately and comprehensively the main advances and limitations of using calcium phosphate-containing biopolymers in the biomedical field and we mentioned the future challenges of these materials. Changing or shorten the text would not preserve the integrity of the semantic message conveyed by the content.

Round 3

Reviewer 2 Report

Nice work and recommended to publish.

Reviewer 3 Report

Although it is a simple review and it is tough to be reproduced, the content is correct and it is satisfactory.

Thank you.